# Life in the fastlane? A comparative analysis of gene expression profiles across annual, semi-annual, and non-annual killifishes (Cyprinodontiformes: Nothobranchiidae)

**Chi Jing Leow¤, Kyle R. Piller** *

Department of Biological Sciences, Southeastern Louisiana University, Hammond, Louisiana, United States of America

¤ Current address: Department of Biochemistry and Cell Biology, Boston University School of Medicine, Boston, Massachusetts, United States of America

* kyle.piller@selu.edu

**Data Availability Statement:** Raw Sequence reads were deposited to NCBI's Sequence Read Archive (BioProject ID: PRJNA1110561). Data can be

## Abstract

The Turquoise Killifish is an important vertebrate for the study of aging and age-related diseases due to its short lifespan. Within Nothobranchiidae, species possess annual, semi-annual, or non-annual life-histories. We took a comparative approach and examined gene expression profiles (QuantSeq) from 62 individuals from eleven nothobranchid species that span three life-histories. Our results show significant differences in differentially expressed genes (DEGs) across life-histories with non-annuals and semi-annuals being most similar, and annuals being the most distinct. At finer scales, we recovered significant differences in DEGs for DNA repair genes and show that non-annual and semi-annuals share similar gene expression profiles, while annuals are distinct. Most of the GO terms enriched in annuals are related to metabolic processes. However, GO terms, including translation, protein transport, and DNA replication initiation also are enriched in annuals. Non-annuals are enriched in Notch signaling pathway genes and downregulated in the canonical Wnt signaling pathway compared to annual species, which suggests that non-annuals have stronger regulation in cellular processes. This study provides support for congruency in DEGs involved in these life-histories and provides strong evidence that a particular set of candidate genes may be worthy of study to investigate their role in the aging process.

## Introduction

Life-history strategies of vertebrates can vary from annualism to multi-year cycles, with the latter being most common [1]. This continuum of longevity is shaped by a complex interplay of ecological and evolutionary factors, which can play a role in the dominant life-history of a particular organism [2]. Much of what is known about longevity in animals comes from mammals. Although there are exceptions, larger mammals generally live longer than small-bodied species [3, 4]. The Bowhead Whale (*Balaena mysticetus*) reaches nearly 20 meters in length

accessed at the following link: https://www.ncbi.
nlm.nih.gov/bioproject/PRJNA1110561.

**Funding:** Research reported in this publication was
supported by an Institutional Development Award
(IDeA) from the National Institute of General
Medical Sciences of the National Institutes of
Health under grant number P2O GM103424-21.
The funders did not play any role in the study
design, data collection and analysis, decision to
publish, or preparation of the manuscript. There
was no additional external funding received for this
study.

**Competing interests:** The authors have declared
that no competing interests exist.

and has a longevity of over 200 years [5]. On the other end of the spectrum, Mueller's Giant
Sundra Rat (*Sundamys muelleri*) reaches a maximum size of 64 cm, and typically lives half a
year in the wild and is the shortest-lived mammal on record [6]. This wide range of lifespans
coupled with comparative studies can offer us insights into the mysteries of longevity and
aging [7].

Within fishes, this scale of longevity also varies dramatically [8, 9]. The large-bodied Green-
land Shark (*Somniosus microcephalus*) recently was shown to have a life-span that reaches over
300 years [10], whereas some short-lived, diminutive fishes, on the opposite end of the size
spectrum, are annual species that complete their entire life-cycle in less than a year. Fishes
with the most reduced life-spans are in the genera *Eviota* (Perciformes: Gobiidae) [11] and
*Nothobranchius* (Cyprinodontiformes: Nothobranchiidae) [12]. In fact, the majority of the
known annual fishes are cyprinodontiforms within the families Nothobranchiidae (African
Rivulines) and Rivulidae (South and Central American Killifishes) [13], two families within
the Suborder Apolocheiloidi that are not sister families, suggesting independent origins of
annual life-histories within the Cyprinodontiformes [14].

Perhaps the most well-known and well-studied annual fish species is the African Turquoise
Killifish (Nothobranchiidae: *Nothobranchius furzeri*), a widespread species that occupies
ephemeral pools in Zimbabwe and Mozambique [15, 16]. The habitats of *N. furzeri* persist for
an average of 75 days during the monsoon season [17], and then begin to desiccate following
the cessation of the rainy season. To survive such harsh and unpredictable environments, *N.
furzeri* possesses an abbreviated life cycle that consists of hatching, rapid growth, sexual matu-
ration, and reproduction in a few weeks. [18], followed by a diapause egg stage to survive
through the dry season [16]. Since the discovery of *N. furzeri* as an annual species, additional
nothobranchid species within *Nothobranchius* and *Pronothobranchius*, have been shown to
have annual life-histories [19].

The extremely short lifespan of *N. furzeri* (and other nothobranchid species) is an appealing
aspect that has made it a model organism in the study of aging and age-related diseases [12].
Historically, model organisms were used for their experimental repeatability and cost of main-
tenance, and when compared to other traditional model organisms like zebrafish and mice,
which have life-spans of three and a half and five years respectively, the GRZ strain of Tur-
quoise Killifish has a lifespan of 10 to 31 weeks in captivity [20]. Additionally, *N. furzeri* shows
a broad set of aging-related dysfunctions including increased risk of cancer [21], locomotor
decline [22], and neurodegeneration [23], making it an appealing model for many age-related
studies in humans. The African Turquoise Killifish genome was independently assembled and
annotated by Reichwald et al. [24] and Valenzano et al. [25]. Since then, the research published
from using *N. furzeri* has grown rapidly. Many other biological disciplines have taken advan-
tage of the rapid life-history of *N. furzeri* in studies spanning evolutionary genomics [24], neu-
rogenesis [26], regenerative biology and medicine [27], developmental biology [28], and
ecotoxicology [29].

In addition to annuals, there are also non-annual and semi-annual genera and species
within Nothobranchiidae. Non-annual species live in non-ephemeral aquatic habitats and they
typically have lifespans of 2–5 years [24, 30]. As a result, non-annual species do not produce
eggs that enter a diapause stage. Within the Nothobranchiidae, non-annual genera include
*Epiplatys*, *Nimbapanchax*, *Archiaphyosemion*, *Scriptaphyosemion*, *Fenerbache*, *Foerschichtys*,
and *Aphyosemion* [31]. On the other hand, the terms "semi-annual" or "facultative-annual"
are used interchangeably in the literature to describe a life cycle that is intermediate between
annual and non-annual species [32]. Semi-annual species produce eggs that are capable of sur-
viving desiccation, but the habitats they live in do not dry out every year [19, 33]. Currently,
*Fundulopanchax* is the only genus that exhibits a semi-annual life history [32].

Biologically, aging is generally defined as the accumulation of damage to somatic cells and DNA [34]. The maintenance of somatic cells, via DNA repair and antioxidant systems, must occur at a high enough level for an organism to survive; once repair and maintenance ability declines below a sustainable level, organisms perish. Promislow [35] noted that there is a strong relationship between DNA repair and longevity such that the capacity of DNA repair in organisms is thought to be related to the array of life-spans observed among organisms. In general, DNA repair mechanisms are highly conserved in animals, levels of expression and particular genes under selection vary according to life-span. MacRae et al. [36] speculated that the lower mutation rates in the Naked Mole Rat was based on a more effective genome mainte-nance mechanism in the mole rats than in mice. To test this hypothesis, they investigated the DNA repair transcriptomes from liver tissues of mammals of varying maximum ages (3–120 years) and expression level differences in the DNA repair related genes. This study showed that the longer-lived species share a higher level of expression of DNA repair genes relative to the short-lived mammals in their study, further substantiating the importance of DNA repair mechanisms to longevity. This study showed that the longer-lived species share a higher level of expression of DNA repair genes relative to the short-lived mammals in their study, further substantiating the potential importance of DNA repair mechanisms to longevity.

The range of life-history variation within the Nothobranchiidae offers the unique opportu-nity to investigate the genetic architecture of three life-histories within a single family and its potential relevance to ageing. In this study, we examined gene expression patterns, across both broad and fine genomic scales, for eleven species of nothobranchid fishes (Cyprinodonti-formes: Nothobranchidae) to test for molecular signatures associated with differences in life-histories (i.e., annuals vs. semi-annuals vs. non-annuals). We hypothesized that the non-annual and semi-annual species will share similar gene expression profiles due to similar life-histories and longevities. We also predicted that the annual species would have the most dis-tinct gene expression profiles relative to the other life-histories, due to their abbreviated and accelerated life cycle. In addition, we took a focused approach and specifically examined the expression patterns of DNA repair genes presented in MacRae et al. [36], to test their potential importance in longevity of nothobranchiid fishes. For DNA repair genes, it was hypothesized that shorter-lived species do not require as much DNA maintenance compared to longer-lived species [36] and that annual species would show downregulation in DNA repair genes com-pared to longer-lived non-annual and semi-annual species.

## Materials and methods

### Specimens and tissue collection

Eleven species of nothobranchids (**Table 1**), across four genera, were obtained from aquarists and hobbyists. Three to six individuals were obtained from each species, and, when possible, equal numbers of males and females from each species were included in this study. To ensure all specimens were harvested at a similar developmental stage (sexually mature), juvenile/sub-adults were kept and raised in tanks until the males displayed nuptial coloration. All notho-branchids are sexually dimorphic and dichromatic, which made it easy to distinguish the sex and maturity of the specimens [37]. When in captivity, fish were fed with frozen bloodworms and maintained at a constant water temperature and light/dark cycle (12hr each), following *N. furzeri* husbandry in research protocol [38] with some modifications. Adult fish were eutha-nized using Tricaine methanesulfonate (MS-222) following the requirement of IACUC stan-dards for fish (American Association for Laboratory Animal Sciences) and "IACUC #0002 at Southeastern Louisiana University. Liver tissues were preserved in RNAlater (Qiagen)

**Table 1. Species of nothobranchid fishes, life-histories, and numbers of males and females included in the study.**

| Species | Abbreviation | Life-History | Males | Females | Total (N) |
|---|---|---|---|---|---|
| *Nothobranchius rubripinnis* | NRb | | 4 | 1 | 5 |
| *Nothobranchius rachovii* | NRv | | 3 | 3 | 6 |
| *Nothobranchius eggersi* | NE | **Annual** | 3 | 3 | 8 |
| *Nothbanchius fuscotaeniatus* | NF | | 4 | 4 | 6 |
| *Fundulopanchax gardneri* | FG | **Semi-Annual** | 3 | 3 | 6 |
| *Epiplatys sexfasciatus* | ES | | 4 | 0 | 4 |
| *Epiplatys guineensis* | EG | | 3 | 0 | 3 |
| *Epiplatys annulatus* | EA | **Non-Annual** | 5 | 1 | 6 |
| *Aphyosemion striatulum* | AS | | 3 | 3 | 6 |
| *Aphyosemion splendopleure* | Asp | | 3 | 3 | 6 |
| *Aphyosemion bivattatum* | AB | | 3 | 3 | 6 |
| Totals = | | | 38 | 24 | 62 |

immediately upon harvesting and stored at -80˚C until RNA isolation. The remaining carcasses were preserved in RNAlater and stored at -80˚C.

## RNA isolation and sequencing

Total RNA was isolated from the liver tissue with RNeasy Plus Universal Mini Kit (Qiagen) following the standard protocol (revised Oct. 2020). The quality of RNA was checked by visualizing the RNA product on a 2% agarose gel. RNA concentrations were measured using Qubit 4 Fluorometer (Thermo Fisher Scientific). Five hundred nanograms of RNA from each individual were prepared and sent to Iowa State University DNA Facility for library prep and sequencing. QuantSeq [39] was chosen over traditional RNA-Seq due to our large sample size and cheaper library preparation cost for QuantSeq. Samples were sequenced using an SP flow cell for 100bp single-end reads on NovaSeq 6000.

## Reads processing and mapping

Raw sequence reads were downloaded and checked with FastQC [40] for quality. Reads were trimmed to remove the first 12bp due to low quality (-u 12), poly-A tail (-n 3 -a "A{20}"), low quality reads (Phred score less than 25), and reads less than 20bp long (-m 20) using cutadapt v.4.1 [41]. Trimmed reads were once again checked with FastQC for quality. The Turquoise Killifish genome (GCF_001465895.1) and the GFF annotation file were downloaded from the National Center for Biotechnology Information (NCBI) website. All reads were aligned to the Turquoise Killifish Genome [25] using STAR 2.7.10a [42]. A genome index was created from the annotated Turquoise Killifish genome for downstream mapping. A 2-pass mapping approach was used as recommended in the STAR manual for a study with multiple samples [42]. The first mapping pass was run for all samples with the default parameters and the junctions were collected. The second mapping pass was run for all samples, listing SJ.out.tab files from all samples generated in the first mapping. The number of reads per gene was counted using–quantMode simultaneously during the second mapping with the following parameters as suggested by QuantSeq mapping protocol --outFilterMultimapNmax 20 --alignSJoverhangMin 8 --alignSJDBoverhangMin 1 --outFilterMismatchNmax 999 --outFilterMismatchNoverLmax 0.6 --alignIntronMin 20 --alignIntronMax 1000000 --alignMatesGapMax 1000000. Read counts of all samples from the third column in the ReadsPerGene.out.tab files were concatenated into an excel file before importing into R [43] and RStudio v.4.2.2. [44] for

analysis and visualization. The datasets generated and analysed during the current study available from the corresponding author.

## Differential gene expression analysis

Low count loci (reads sum < 10) were filtered out to reduce the memory size of the data object and increase the speed of downstream analyses. Size factors were taken into account for different library sizes. Differential gene expression (DGE) analyses were conducted using the R package DESeq2 v.1.36.0 [45]. Differential gene expression analyses were conducted in pairwise comparison between life histories: annual vs. non-annual (A vs. N), annual vs. semi-annual (A vs. SA), and semi-annual vs. non-annual (SA vs. N). The false discovery rate cutoff (alpha) was set to 0.05 and the log2FoldChange was set to 1 (a two-fold change) to detect DEGs.

We took a two-pronged approach to examine DGE across life-histories. First, we examined the 500 most variable genes to give a broad perspective of DGE across life-histories using Principal Component Analysis (PCA) in DESeq2 [45]. Next, we specifically examined the 130 DNA repair genes identified by MacRae et al. [36] to test for conservation of expression patterns of these genes across life-histories. This focused analysis is based on the idea that shorter-lived species do not require as much DNA maintenance compared to longer-lived species, as was observed in a set of mammals with various longevities by MacRae et al [36]. As a result, it is expected that annual, semi-annual, and non-annual nothobranchids will show different gene expression patterns of these genes. Volcano plots of the DEGs (pairwise comparisons between life histories) were made using the function *produce volcano* from a R package rnaseq v.1.0.8. [46].

## Gene ontology enrichment and KEGG pathway analyses

All the significant differentially expressed genes (DEGs) were separated into an upregulated gene list and downregulated gene list from each pairwise comparison. Hong et al. [47] noted that analyzing up- and downregulated genes separately is more powerful to detect meaningful GO terms than analyzing all DEGs simultaneously. These gene lists were then uploaded separately to the DAVID 2021 webserver [48] for GO term enrichment analysis and KEGG pathway analysis. When necessary, genes submitted to the DAVID webserver were programmatically converted from official gene symbols to entrez gene IDs by DAVID. Gene IDs that were not convertible in the DAVID database were filtered out and the remaining genes were submitted to DAVID with *N. furzeri* as background (**S1 Table**). An EASE Score of 0.05, a modified Fisher Exact *p*-value for gene enrichment analysis, and a gene count of 2 (default by DAVID) were used to determine significantly enriched GO terms and KEGG pathways. The GO terms from biological process (BP) direct and the associated p-values were uploaded to REVIGO [49] to summarize them by removing redundant GO terms. All GO terms and the associated p values were run through REVIGO with the same settings: removed obsolete GO terms, searched against the Whole UniProt database (Turquoise Killifish was not an option), and used SimRel as sematic similarity measure.

# Results and discussion

## Sequence reads

Over 627 million reads were obtained from the 62 individuals, averaging 10.12 million reads per sample. Samples NRbf1 and EG4 were discarded due to low quality reads. After trimming with cutadapt [41], the remaining 591.9 million reads were mapped to the *N. furzeri* genome with STAR using 2-pass mode [42]. Raw Sequence reads were deposited to NCBI's Sequence

Read Archive (BioProject ID: PRJNA1110561). Scripts and datasets used in this study are archived at https://zenodo.org/records/13213974 (DOI 10.5281/zenodo.13213973).

## Gene expression profiles

Principal Component Analysis (PCA), based on the 500 (default by DESeq2) most variable genes from the QuantSeq data, was conducted to examine variation in the species, genera, and life-histories in multivariate space (Fig 1). The analysis recovered three non-overlapping clusters, with the first two PC axes accounting for 58% of the total variance (PC1 44% and PC2 14%). One cluster consisted of four annual species, *N. rachovii*, *N. rubripinnis*, *N. eggersi*, and *N. fuscotaeniatus*, that was separated from both semi-annual and non-annual clusters along PC1. A second cluster consisted solely of non-annual species of the genus *Epiplatys* (*E. sexfasciatus*, *E. guinneensis*, and *E. annualatus*). Finally, the third cluster consisted of semi-annual (*F. gardneri*) and non-annual (*A. striatum*, *A. splendopleure*, *A. bivittatum*) species. Overall, at this level, these results suggest that phylogenetic constraint is a strong driver of the recovered clusters.

## Differentially expressed genes

The gene expression patterns were compared across life-histories with a two-fold change in expression level and a false discovery rate (FDR) below 0.05. There were 4,551 DEGs found between annuals and non-annuals (Fig 2A). Among the 4,551 DEGs, 3,731 genes are upregulated and 820 are downregulated in annuals compared to non-annuals. (Fig 2B). There are 2,345 DEGs found between annuals and semi-annuals with 2,002 upregulated genes and 343 downregulated genes. Finally, there are 1,187 DEGs found between semi-annuals and non-

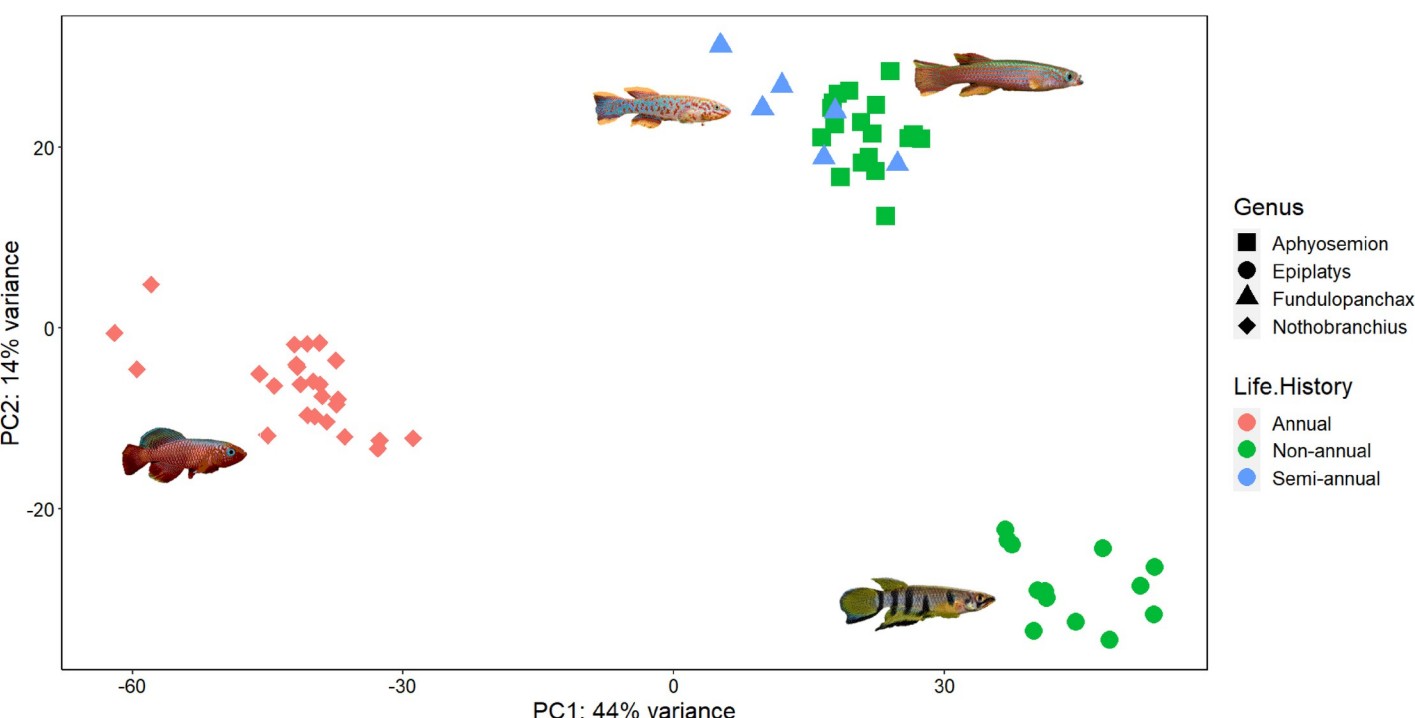

**Fig 1. Principal component analysis (PCA) of 62 individuals based on the top 500 most variable expressed genes (QuantSeq).** The genera are designated by shape, and the life-histories are designated by colors.

### A. Annuals vs. Non-annuals

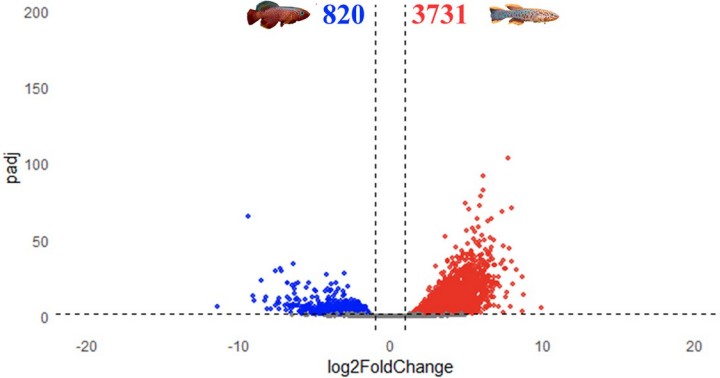

### B. Annuals vs. Semi-annuals

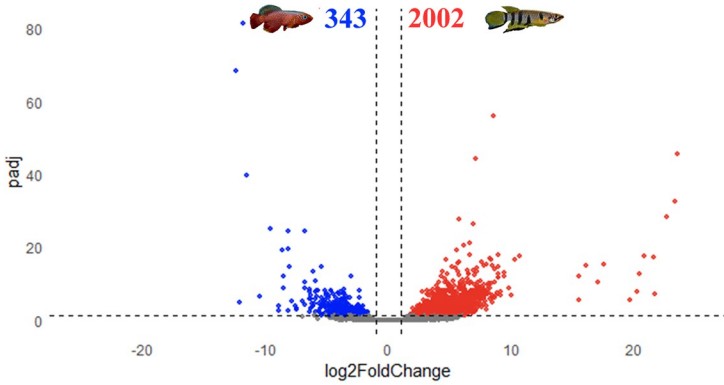

### C. Semi-annuals vs. non-annuals

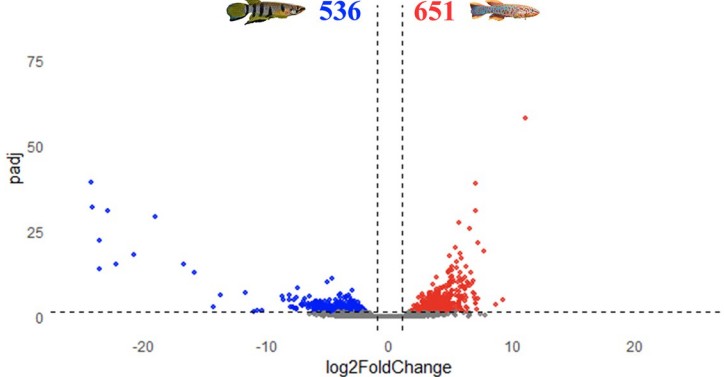

**Fig 2. Volcano plots of the differential expressed genes between life-histories.** A). annuals vs. non-annuals, B). annuals vs. semi-annuals, and C). semi-annuals vs. non-annuals. Red represents significant upregulated genes. Blue represents significant downregulated genes.

annuals with 651 upregulated and 536 downregulated genes (**Fig 2C**). These results highlight the distinct DEG patterns of annuals in comparison to semi-annuals and non-annuals.

## Gene ontology enrichment analysis and visualization

We also conducted gene ontology (GO) enrichment analysis using the DAVID Bioinformatics Resources webserver [48] to understand what biological processes might be affected by these

DEGs. We found 14 enriched GO biological process terms in annuals compared to non-annuals (**Fig 3A–3D**) and five enriched GO biological process terms in annuals compared to semi-annuals. We found three enriched GO biological process terms in non-annuals compared to annuals. Finally, only one GO biological process term, is enriched in non-annuals compared to semi-annuals.

Gene Ontology (GO) terms that are enriched in annuals compared to non-annuals include: protein transport, carbohydrate metabolic process, mitochondrial translation, fatty acid metabolic process, defense response to bacterium, spliceosomal snRNP assembly, vesicle-mediated transport, tRNA 5'-leader removal, isoprenoid biosynthetic process, translation, cholesterol biosynthetic process, ER to Golgi vesicle-mediated transport, DNA replication initiation, and

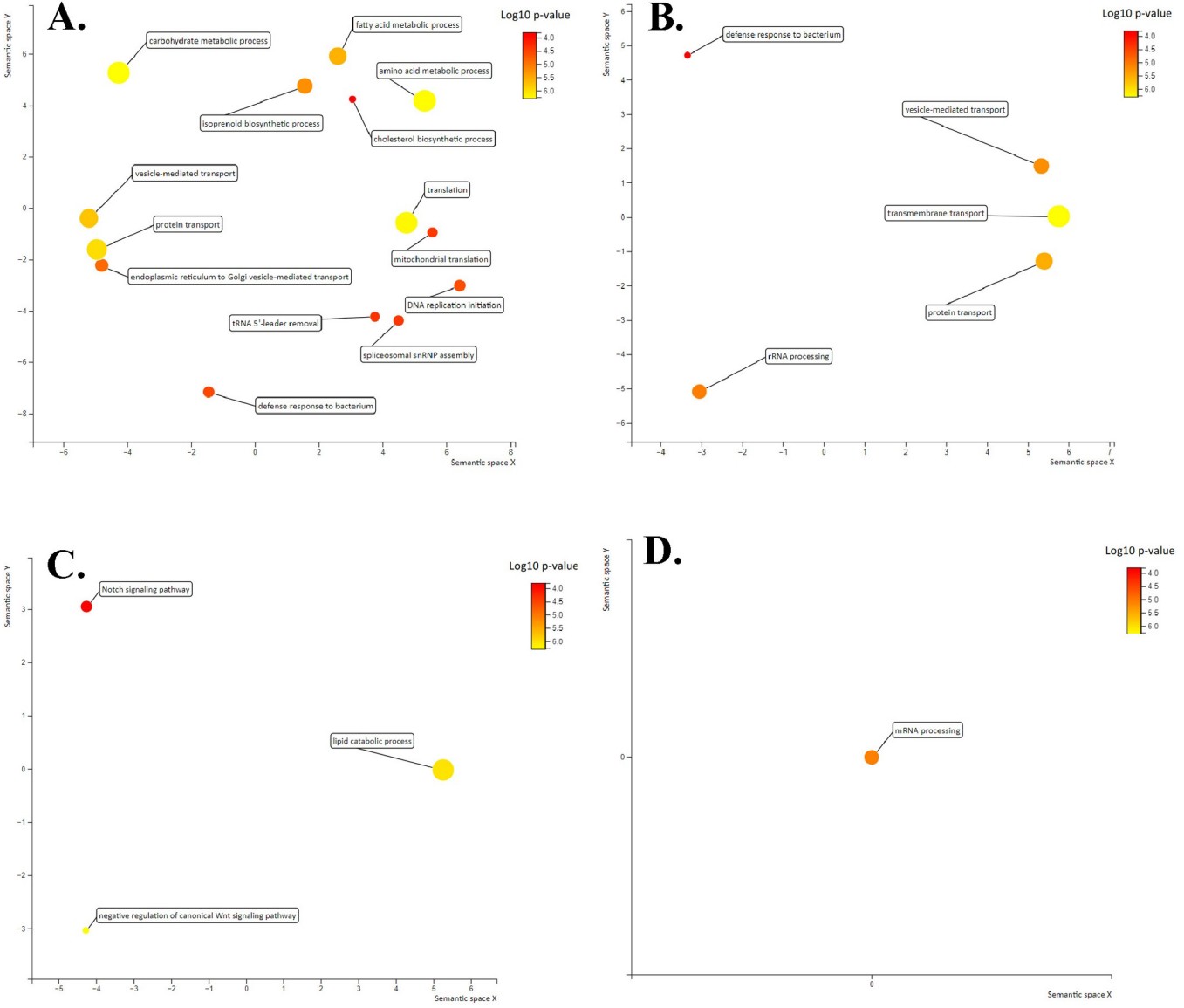

**Fig 3.** Enriched biological processes GO terms in A). annual species compared to non-annuals, B). annual species compared to semi-annual species, C). non-annual species compared to annual species, and D) non-annual species compared to semi-annual species visualized in an MDS plot. X and Y axis are sematic similarity which circles cluster together are more closely related. Circle color and size indicate log10 p-value of each GO term. The threshold of EASE Score, a modified Fisher Exact *p*-value for gene enrichment analysis was set to 0.05.

cellular amino acid metabolic process (**Fig 3A**). Gene Ontology terms that are enriched in annuals compared to semi-annuals include: protein transport, rRNA processing, vesicle-mediated transport, transmembrane transport, and defense response to bacterium (**Fig 3B**). Gene Ontology terms that are significantly enriched in non-annuals compared to annuals are Notch signaling pathway, negative regulation of canonical Wnt signaling pathway and lipid catabolic processes (**Fig 3C**). The only significantly enriched GO biological process term found in non-annuals compared to semi-annuals is mRNA processing (**Fig 3D**). Similar to the previous analyses, the GO term analyses highlight the distinctive transcriptomic signature of annual fishes relative to the life-histories.

## KEGG pathway analysis

From the DEGs, we also conducted KEGG pathway analysis using the DAVID Bioinformatics Resources webserver [48]. Annual species are enriched in 38 and 22 KEGG pathways compared to non-annual and semi-annual species respectively (**S2 and S3 Tables**). Most of these KEGG pathways enriched in annual species are related to metabolism. Some of the non-metabolic related pathways enriched in annual species include DNA replication and cell cycle pathways. Semi-annual species are enriched in 11 KEGG pathways compared to non-annual species in which all of them are related to metabolism (**S4 Table**). Non-annual species are enriched in eight KEGG pathways including Notch and Wnt signaling pathways compared to annual species (**S5 Table**). Semi-annual species are enriched in five KEGG pathways compared to annual species in which all of them are related to metabolism (**S6 Table**). Lastly, non-annual species are enriched in only two KEGG pathways compared to semi-annual species including: phagosome and endocytosis (**S7 Table**).

## DNA repair genes

MacRae et al. [36] examined differences in gene expression patterns from liver tissue for three mammals (humans, naked mole rate, and mice) across divergent maximum lifespans (120, 30, and 3 years, respectively). Their results show different expression patterns for DNA repair genes from liver tissue. We specifically examined the expression levels of the 130 DNA repair genes identified by MacRae et al. [36] in our QuantSeq dataset. Thirty-one of the 130 DNA repair genes are differentially expressed between annuals and non-annuals, with XRCC3, GTF25, and PMS1 having the highest log2foldchanges (**Table 2 and Fig 4**). Among those 31 genes, 27 are upregulated and four are downregulated in annuals. Fifteen DNA repair genes are differentially expressed between annuals and semi-annuals in which 14 are upregulated and only one downregulated (**Table 3**). Only three DNA repair genes are differentially expressed between semi-annuals and non-annuals in which two are upregulated in semi-annuals (**Table 4**). These 36 differentially expressed DNA repair genes from life-history comparisons (annuals vs. non-annuals, annuals vs. semi-annuals, and semi-annuals vs. non-annuals) are depicted in a heatmap (**Fig 5**).

## Gene expression profiles

Comparative transcriptomics is a powerful tool for studying the genetic architecture attributable to unique life-histories, differences in lifespans, and convergent evolution in a wide range of taxa [36, 50, 51]. In our study, the broad gene expression profiles of nothobranchid fishes are clustered by phylogeny rather than life-history. Contrary to our hypothesis and to the mammal study of MacRae et al. [36], shorter-lived species (annuals) are upregulated in many DNA repair genes compared to longer-lived species (semi-annuals and non-annuals). Most of the significantly enriched GO terms and KEGG pathways in annual species are related to

**Table 2. Significant differential expressed DNA repair genes in annuals compared to non-annuals.** Genes are ranked from highest log2foldchange to lowest. There are 27 upregulated and 4 downregulated DNA repair genes. Differential expression was determined by having a log2foldchange greater than one and padj value of less than 0.05. Base mean is the mean of normalized counts for all samples. LfcSE is the standard error. Stat is Wald statistics. Pvalue is the Wald test p-value. Padj is the Benjamini-Hochberg adjusted p-value.

| Gene symbol | baseMean | log2FoldChange | lfcSE | stat | pvalue | Padj |
|---|---|---|---|---|---|---|
| XRCC3 | 8.299 | 5.629 | 0.604 | 7.665 | 1.79E-14 | 3.91E-13 |
| GTF2H5 | 18.907 | 5.094 | 0.405 | 10.105 | 5.27E-24 | 3.05E-22 |
| PMS1 | 25.701 | 4.478 | 0.636 | 5.468 | 4.56E-08 | 4.55E-07 |
| RPA1 | 16.333 | 3.766 | 0.327 | 8.461 | 2.65E-17 | 7.67E-16 |
| ALKBH3 | 3.008 | 3.554 | 0.426 | 5.994 | 2.05E-09 | 2.48E-08 |
| EXO1 | 1.324 | 3.534 | 0.674 | 3.761 | 1.69E-04 | 9.45E-04 |
| UNG | 1.779 | 3.509 | 0.506 | 4.96 | 7.07E-07 | 5.87E-06 |
| FANCC | 3.252 | 3.502 | 0.548 | 4.563 | 5.05E-06 | 3.67E-05 |
| MAD2L2 | 2.299 | 3.297 | 0.558 | 4.116 | 3.86E-05 | 2.41E-04 |
| POLM | 17.441 | 3.178 | 0.436 | 4.993 | 5.95E-07 | 5.02E-06 |
| PCNA | 16.459 | 3.106 | 0.361 | 5.826 | 5.67E-09 | 6.51E-08 |
| BRCA1 | 6.191 | 3.027 | 0.34 | 5.967 | 2.42E-09 | 2.91E-08 |
| BRCA2 | 15.872 | 2.914 | 0.335 | 5.723 | 1.05E-08 | 1.15E-07 |
| DCLRE1B | 2.559 | 2.854 | 0.603 | 3.073 | 2.12E-03 | 9.27E-03 |
| DCLRE1C | 5.561 | 2.61 | 0.316 | 5.096 | 3.47E-07 | 3.03E-06 |
| BRIP1 | 1.628 | 2.558 | 0.613 | 2.541 | 1.11E-02 | 4.10E-02 |
| TP53 | 28.11 | 2.395 | 0.383 | 3.644 | 2.68E-04 | 1.43E-03 |
| RNF4 | 6.447 | 2.366 | 0.52 | 2.628 | 8.58E-03 | 3.27E-02 |
| FAN1 | 2.909 | 2.33 | 0.388 | 3.43 | 6.03E-04 | 2.99E-03 |
| POLI | 3.375 | 2.32 | 0.491 | 2.691 | 7.13E-03 | 2.79E-02 |
| XRCC6 | 12.955 | 2.147 | 0.441 | 2.603 | 9.25E-03 | 3.50E-02 |
| LIG4 | 5.982 | 2.09 | 0.375 | 2.904 | 3.69E-03 | 1.54E-02 |
| GTF2H4 | 33.041 | 2.09 | 0.292 | 3.728 | 1.93E-04 | 1.07E-03 |
| APLF | 11.916 | 2.063 | 0.297 | 3.574 | 3.52E-04 | 1.83E-03 |
| WRN | 6.991 | 2.047 | 0.286 | 3.656 | 2.56E-04 | 1.37E-03 |
| SHPRH | 19.854 | 1.534 | 0.198 | 2.701 | 6.91E-03 | 2.71E-02 |
| COPS5 | 23.8 | 1.499 | 0.202 | 2.467 | 1.36E-02 | 4.92E-02 |
| HELQ | 1.245 | -2.677 | 0.669 | -2.504 | 1.23E-02 | 4.49E-02 |
| CLK2 | 85.826 | -2.981 | 0.292 | -6.773 | 1.26E-11 | 1.99E-10 |
| POLL | 11.262 | -3.129 | 0.342 | -6.228 | 4.71E-10 | 6.20E-09 |
| APEX2 | 1.587 | -3.175 | 0.643 | -3.383 | 7.18E-04 | 3.49E-03 |

metabolic processes with exceptions such as DNA replication and cell cycle suggesting an accelerated life cycle in the annual species. Alternatively, longer-lived species (non-annuals) are enriched in Notch signaling pathway and negative regulation of Wnt signaling pathway compared to annual species suggesting that longer-lived species have better regulation in cellular processes including cell proliferation, cell death, and homeostasis.

The three distinct clusters show strong phylogenetic signal (**Fig 1**). The four *Nothobranchius* species and the three species of *Epiplatys* each form distinct clusters. The third cluster includes *Aphyosemion*, which are non-annual species, and one species of *Fundulopanchax*, a semi-annual genus. *Aphyosemion* and *Fundulopanchax* are sister groups based on a previous phylogenetic study [52], which suggests that *Aphyosemion* and *Fundulopanchax* may be clustered because of their close phylogenetic affinities rather than their life-histories, when overall gene expression patterns are examined. Relative to the *Aphyosemion/Fundulopanchax* cluster,

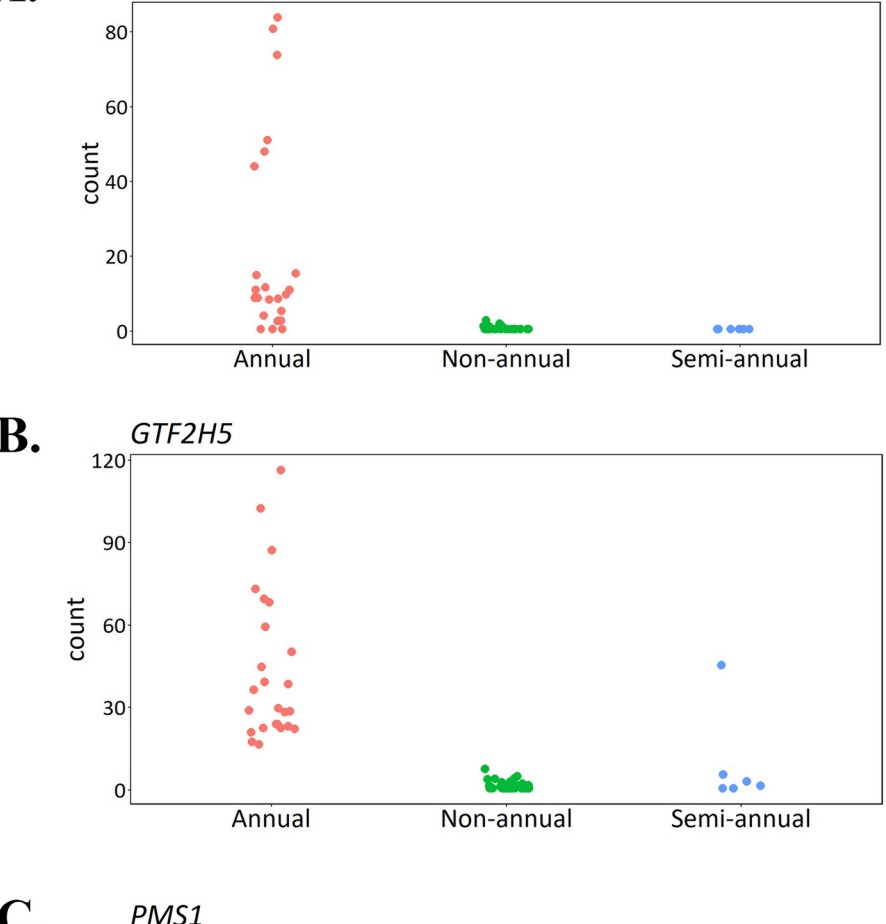

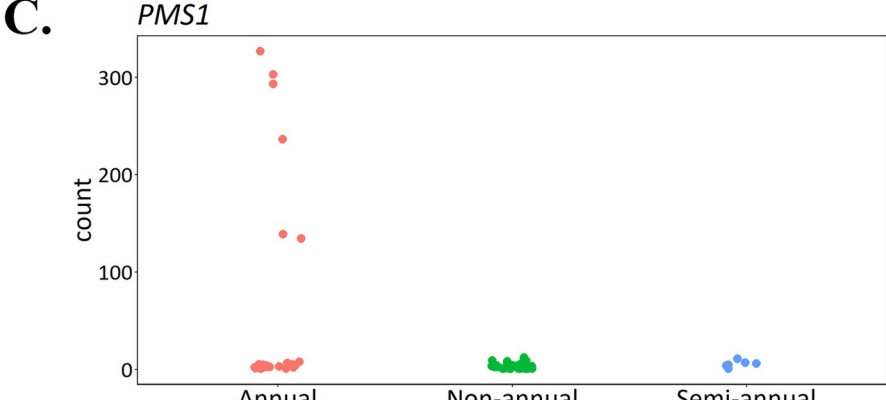

**Fig 4.** The top three most differentially expressed DNA repair genes (annuals vs. non-annuals) based on log2foldchange values: A). GTF25, B). PMS1, and C) XRCC3.

the other two clusters are phylogenetically distant from one another and from the *Aphyosemion/Fundulopanchax* clade [52]. The gene expression data from these killifish clustered more by their phylogenetic relationships rather than life history variation. Within clusters, species likely share many DEGs that are genus-specific rather than related to life history and this is further supported by the fact that we found that many of these highly variable DEGs are uncharacterized loci with orthologs that have not yet been identified (**S8 Table**).

**Table 3. Significant differential expressed DNA repair genes in annuals compared to semi-annuals.** Genes are ranked from highest log2foldchange to lowest. There are 14 upregulated and one downregulated DNA repair genes. Differential expression was determined by having a log2foldchange greater than one and padj value of less than 0.05. Base mean is the mean of normalized counts for all samples. LfcSE is the standard error. Stat is Wald statistics. Pvalue is the Wald test p-value. Padj is the Benjamini-Hochberg adjusted p-value.

| Gene symbol | baseMean | log2FoldChange | lfcSE | stat | pvalue | Padj |
|---|---|---|---|---|---|---|
| GTF2H4 | 33.041 | 7.171 | 0.959 | 6.437 | 1.22E-10 | 1.16E-08 |
| XRCC3 | 8.299 | 6.024 | 1.166 | 4.309 | 1.64E-05 | 2.97E-04 |
| DCLRE1C | 5.561 | 5.233 | 0.921 | 4.595 | 4.33E-06 | 1.02E-04 |
| POLM | 17.441 | 4.962 | 0.961 | 4.125 | 3.72E-05 | 5.93E-04 |
| PCNA | 16.459 | 4.818 | 0.858 | 4.448 | 8.65E-06 | 1.77E-04 |
| ALKBH3 | 3.008 | 4.473 | 0.933 | 3.721 | 1.98E-04 | 2.40E-03 |
| RAD17 | 3.298 | 4.347 | 1.097 | 3.052 | 2.27E-03 | 1.76E-02 |
| DCLRE1B | 2.559 | 4.149 | 1.195 | 2.636 | 8.39E-03 | 4.92E-02 |
| GPS1 | 5.784 | 3.66 | 0.888 | 2.996 | 2.73E-03 | 2.04E-02 |
| BRCA1 | 6.191 | 3.373 | 0.769 | 3.087 | 2.02E-03 | 1.60E-02 |
| XAB2 | 6.938 | 3.343 | 0.843 | 2.78 | 5.44E-03 | 3.49E-02 |
| POLB | 8.783 | 3.272 | 0.774 | 2.937 | 3.31E-03 | 2.37E-02 |
| TP53 | 28.11 | 3.077 | 0.699 | 2.973 | 2.95E-03 | 2.17E-02 |
| BRCA2 | 15.872 | 2.805 | 0.623 | 2.898 | 3.76E-03 | 2.62E-02 |
| GTF2H5 | 18.907 | 2.628 | 0.608 | 2.679 | 7.39E-03 | 4.45E-02 |
| POLL | 11.262 | -3.196 | 0.553 | -3.972 | 7.11E-05 | 1.01E-03 |

## Global deregulation of genes

From the DGE analysis, annual species have a large number of DEGs compared to non-annual and semi-annual species, with the smallest number of DEGs between non-annual and semi-annual species, suggesting that annual species have more DEGs in the liver during the early adult stage compared to the other life-histories. A recent study conducted by Fumagalli et al. [53] found that *N. furzeri* has a higher percentage of deregulated genes (45%) during aging, when compared to the zebrafish (*Danio Rerio*) and mouse (*Mus musculus*). These DEGs in *N. furzeri* during aging are randomly distributed throughout the genome, rather than displaying a tissue and time-specific regulation in the mouse [53]. Similarly, Barth et al. [54] compared the gene expression patterns of different tissues in four species (humans, mouse, zebrafish, and turquoise killifish) at different time points during aging and found that long-lived individuals show tighter regulation in genes related to aging compared to short-lived individuals. Our results suggest a similar pattern where short-lived annual species display more differentially expressed genes compared to longer-lived non-annual and semi-annual species.

**Table 4. Significant differential expressed DNA repair genes in semi-annuals compared to non-annuals.** Genes are ranked from highest log2foldchange to lowest. There are two upregulated and one downregulated DNA repair genes. Differential expression was determined by having a log2foldchange greater than one and padj value of less than 0.05. Base mean is the mean of normalized counts for all samples. LfcSE is the standard error. Stat is Wald statistics. Pvalue is the Wald test p-value. Padj is the Benjamini-Hochberg adjusted p-value.

| Gene symbol | baseMean | log2FoldChange | lfcSE | stat | pvalue | Padj |
|---|---|---|---|---|---|---|
| NUDT1 | 2.138 | 3.775 | 0.904 | 3.07 | 0.00214 | 0.0309 |
| RPA3 | 19.662 | 3.552 | 0.686 | 3.718 | 0.00020 | 0.0051 |
| GTF2H4 | 33.041 | -5.081 | 0.957 | -4.265 | 0.00002 | 0.0008 |

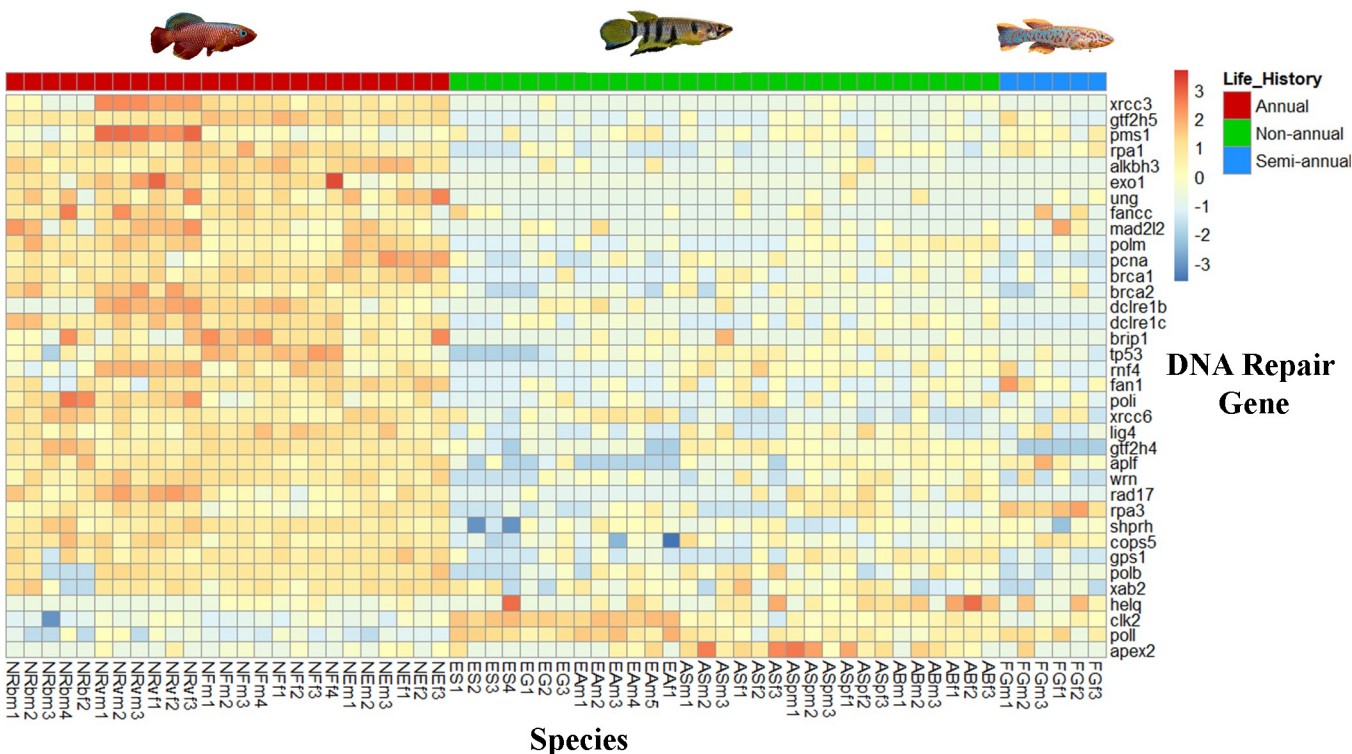

**Fig 5. Heatmap of the significant differentially expressed DNA repair genes between life-histories.** Genes are ranked from highest to lowest expression based on the log2fold change. Column colors represent life-histories (annuals: green, red: non-annuals, blue: semi-annuals).

### Upregulation of DNA repair genes

In terms of longevity, DNA repair plays a crucial role. Many factors contribute to the process of aging and genomic instability is one of the nine hallmarks of aging [55]. The integrity of DNA is constantly being challenged by exogenous physical and chemical agents, as well as endogenous molecules that damage DNA [56]. The expression levels of DNA repair genes are directly related to the longevities of organisms [36]. In our study, many of the DNA repair genes in annuals were upregulated in comparison to the long-lived species (non-annuals and semi-annuals). This is contrary to what MacRae et al. [36] presented in their study of three mammal species of varying ages, where longer-lived species exhibited higher expression levels in DNA repair genes and pathways. Studies have shown that gene expression itself is affected by aging [57, 58] and the dysregulation of gene expression and mRNA processing is one of the hallmarks of cellular aging in eukaryotes [59]. Therefore, it is likely that the many DEGs in annuals could be attributed to the dysregulation of genes during aging.

Another possible explanation for the upregulation of DNA repair genes in annual killifish is that they have accumulated more deleterious gene variants compared to non-annual species [60]. There is a strong selection of genes related to rapid maturation in the annual killifish in order to reproduce in the ephemeral habitat [25]. The genetic drift in annual species has led to expansion of nuclear and mitochondrial genomes which causes accumulation of deleterious variants particularly in aging-related and DNA repair genes [60, 61]. This increase of genome-wide mutation load is proposed to be a key factor in the accelerated aging in annual killifish by affecting the adult life fitness [60, 61]. Mutations in DNA repair genes could cause the DNA repair mechanisms to function improperly and inefficiently. Therefore, the upregulation of

DNA repair genes might be necessary for the annual killifish to maintain the required levels of DNA repair mechanisms for survival.

## Gene ontology enrichment analysis

As expected from liver tissue, most of these GO terms enriched in annuals are related to metabolism. Other than the metabolic process GO terms, annual species are also upregulated in GO terms including translation, protein transport, and DNA replication initiation. These GO terms are likely derived from the annual life-history rather than related to liver functions. Studies have shown that the upregulation of translation and ribosomal processes are indicators of aging in *N. furzeri* [24, 62]. The upregulation of DNA replication initiation in annuals compared to non-annuals likely results from their rapid life cycle.

We did not discover GO terms related to DNA damage and DNA repair in the GO enrichment analysis. The 31 differentially expressed DNA repair genes are small in number when compared to more than 4,000 DEGs between annuals and non-annuals. MacRae et al. [36] only detected DNA repair related GO terms from signaling pathway analysis but not in their functional annotation analysis. MacRae et al. [36] concluded that it was difficult to detect the small changes in DNA repair genes with GO enrichment analysis when there are large numbers of DEGs related to metabolism found in the liver. It is likely that the GO enrichment analysis we used was not able to detect these subtle differences in expressions for DNA repair functions, when there were only 31 DNA repair genes that showed significant differential expression compared to the other 4,000 DEGs.

## KEGG pathway analysis

Kyoto Encyclopedia of Genes and Genomes (KEGG) is an open-source database that assigns functional meanings to genes and genomes [63]. We conducted KEGG pathway analysis of the DEGs using the DAVID webserver [48]. As expected from the liver tissue, annual species are enriched in many pathways related to metabolism compared to other life-histories (**S2 and S3 Tables**). However, it was previously shown that *N. furzeri* exhibits a rapid increase in metabolic activity during its lifetime compared to zebrafish and mouse, which have longer lifespans [53]. Compared to non-annual species, annual species are also enriched in DNA replication and translation pathways which suggests an accelerated life cycle [25]. Furthermore, these enriched pathways in annual species are what would be expected for a species with accelerated growth in the wild and captivity due to strong selection on genes related to early life stages including embryonic development and sexual maturation [12, 25, 60].

Non-annual species are only enriched in negative regulation of the canonical Wnt signaling pathway, Notch signaling pathway, and lipid catabolic process (**S3 Table**) compared to annual species. The canonical Wnt signaling pathway is a conserved pathway in metazoan animals, which is involved in many events during embryonic development such as cell fate determination, cell migration, cell polarity, neural patterning, and organogenesis [64]. The negative regulation of the canonical Wnt signaling pathway refers to any process that reduces the rate and duration of Wnt signaling pathway in the cells [65]. The deregulation of the Wnt signaling pathway previously has been shown to contribute to various hepatic pathologies [66] and colorectal cancer in humans [67]. Willemsen et al. [61] also discovered that short-lived populations of *N. furzeri* have higher mutations in genes related to Wnt signaling pathways compared to longer-lived populations. Our results show that non-annual species are downregulating the Wnt signaling pathway compared to annual species. This could suggest that the longer-lived non-annual species do not need the same levels of Wnt signaling pathway in the liver tissue compared to annual species.

Non-annual species are also enriched in Notch signaling pathway compared to annual species. Notch is an ancient and highly conserved signaling pathway in all metazoans [68, 69]. The Notch pathway regulates many important cellular mechanisms including cell proliferation, cell fate, differentiation, and cell death [69]. Many recent studies have highlighted the role of Notch signaling pathway in aging such as maintaining homeostasis and tissue specific homeostasis [70–72]. It is shown that the level of Notch activity is negatively correlated with aging [73]. Not only that, the deregulation of Notch signaling pathway has also been associated with cancers [74]. This study shows that annual species are downregulated in Notch signaling pathway compared to non-annual species, which suggests that annual species have lower regulation in cellular processes including cell proliferation, cell death, and homeostasis due to their rapid life cycle.

## Conclusions

The process of aging has been extensively studied and model organisms have long been fused to study and understand the mechanisms behind it. The use of non-canonical model organisms, such as the African Turquoise Killifish, has provided many new insights in the study of aging and age-related diseases [75, 76]. Here, we investigated the gene expression profiles of closely related nothobranchid killifishes with different life-histories. By sequencing the mRNA from liver tissues and conducting DGE analyses at a broad level (PCA), as well as examining DNA repair genes, we were able to gain a better picture of the gene expression profiles of these killifishes and show that non-annual species and semi-annual species share more similar gene expression profiles, while annual species are the most distinct group.

*Nothobranchius furzeri* has become a widely used model organism in many biological areas due to its rapid generation time which make it amenable and convenient for many studies [24, 26–29]. Methods for transgenesis and genome engineering have been previously reported for *N. furzeri* [77, 78] and protocols for CRISPR-Cas9 genome editing in *N. furzeri* have recently been published [79]. Many of the upregulated and downregulated genes across life-histories in our study could be used in future transgenic and gene silencing studies involving *N. furzeri* to examine their impacts on longevity in that model species. Due to the high degree of homology between *N. furzeri* and human genomes [25], our results could have strong implications on augmenting our understanding of aging and age-related diseases in humans.

## Supporting information

**S1 Table. Number of DEGs submitted and filtered out in the DAVID webserver.**
(DOCX)

**S2 Table. KEGG: Annuals vs. Non-annuals (liver).** Enriched pathways obtained from submitting the DEGs to DAVID webserver. Threshold of minimum gene counts 2 (belonging to an annotation term) and EASE score threshold 0.05 were used to determine significant KEGG pathways.
(DOCX)

**S3 Table. KEGG: Annuals vs. Semi-annuals (liver).** Enriched pathways obtained from submitting the DEGs to DAVID webserver. Threshold of minimum gene counts 2 (belonging to an annotation term) and EASE score threshold 0.05 were used to determine significant KEGG pathways.
(DOCX)

**S4 Table. KEGG: Semi-annuals vs. Non-annuals (liver).** Enriched pathways obtained from submitting the DEGs to DAVID webserver. Threshold of minimum gene counts 2 (belonging to an annotation term) and EASE score threshold 0.05 were used to determine significant

KEGG pathways.
(DOCX)

**S5 Table. KEGG: Non-annuals vs. Annuals (liver).** Enriched pathways obtained from submitting the DEGs to DAVID webserver. Threshold of minimum gene counts 2 (belonging to an annotation term) and EASE score threshold 0.05 were used to determine significant KEGG pathways.
(DOCX)

**S6 Table. KEGG: Semi-annuals vs. Annuals (liver).** Enriched pathways obtained from submitting the DEGs to DAVID webserver. Threshold of minimum gene counts 2 (belonging to an annotation term) and EASE score threshold 0.05 were used to determine significant KEGG pathways.
(DOCX)

**S7 Table. KEGG: Non-annuals vs. Semi-annuals (liver).** Enriched pathways obtained from submitting the DEGs to DAVID webserver. Threshold of minimum gene counts 2 (belonging to an annotation term) and EASE score threshold 0.05 were used to determine significant KEGG pathways.
(DOCX)

**S8 Table. The top 500 most significant differentially expressed genes in annuals vs. Non-annuals.** Genes are ranked by the lowest padj value.
(DOCX)

## Acknowledgments

We would like to thank Brant Faircloth (Louisiana State University), Mary White (Southeastern Louisiana University), and April Wright (Southeastern Louisiana University), who provided constructive criticism on an earlier version of this manuscript. Portions of this research were conducted with high performance computational resources provided by the Louisiana Optical Network Infrastructure (http://www.loni.org).

## Author Contributions

**Conceptualization:** Kyle R. Piller.

**Data curation:** Chi Jing Leow.

**Formal analysis:** Chi Jing Leow.

**Funding acquisition:** Kyle R. Piller.

**Investigation:** Chi Jing Leow.

**Methodology:** Chi Jing Leow.

**Project administration:** Kyle R. Piller.

**Resources:** Kyle R. Piller.

**Supervision:** Kyle R. Piller.

**Visualization:** Chi Jing Leow.

**Writing – original draft:** Chi Jing Leow, Kyle R. Piller.

**Writing – review & editing:** Chi Jing Leow, Kyle R. Piller.

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
