## [Decision Letter · Decision Letter 0]

28 Feb 2024

PONE-D-23-41180Life in the fastlane? A comparative analysis of gene expression profiles across annual, semi-annual, and non-annual killifishes (Cyprinodontiformes:Nothobranchiidae)PLOS ONE

Dear Dr. Piller,

Thank you for submitting your manuscript to PLOS ONE. After careful consideration, we feel that it has merit but does not fully meet PLOS ONE’s publication criteria as it currently stands. Therefore, we invite you to submit a revised version of the manuscript that addresses the points raised during the review process.

We look forward to receiving your revised manuscript.

Kind regards,

Livia D'Angelo, Ph.D

Academic Editor

PLOS ONE

Journal Requirements:

KRP - Research reported in this publication was supported by an Institutional Development Award (IDeA) from the National Institute of General Medical Sciences of the National Institutes of Health under grant number P2O GM103424-21

The funders did not play any role in the study design, data collection and analysis, decision to publish, or preparation of the manuscript.

This study was supported by funding to KRP from the Louisiana Biomedical Research Network. Specifically, the research reported in this publication was supported by an Institutional Development Award (IDeA) from the National Institute of General Medical Sciences of the National Institutes of Health under grant number P2O GM103424-21.

KRP - Research reported in this publication was supported by an Institutional Development Award (IDeA) from the National Institute of General Medical Sciences of the National Institutes of Health under grant number P2O GM103424-21

The funders did not play any role in the study design, data collection and analysis, decision to publish, or preparation of the manuscript.

Reviewers' comments:

Reviewer's Responses to Questions

**Comments to the Author**

1. Is the manuscript technically sound, and do the data support the conclusions?

Reviewer #1: Yes

2. Has the statistical analysis been performed appropriately and rigorously? 

Reviewer #1: Yes

3. Have the authors made all data underlying the findings in their manuscript fully available?

Reviewer #1: Yes

4. Is the manuscript presented in an intelligible fashion and written in standard English?

Reviewer #1: Yes

5. Review Comments to the Author

Reviewer #1: The manuscript submitted by Leow and Piller is a descriptive genomic analysis of gene expression profiles from 62 individuals belonging to eleven different Nothobranchiidae species that fall into three different life-histories (annual, semi-annual, and non-annual). The work provides some new information on how gene regulation can change with life-history, with the possibility of studying the genes highlighted in the manuscript in transgenic and gene silencing studies and examining their impact on longevity. However, it shows also some flaws, the first one being the lack of a deeper discussion on some of the main results, such as the Gene Ontology analysis, as well as the regulation of DNA repair genes. I think that this could be achieved in different ways: creating some heatmaps or pathway visualization to take a further look at the pathways involved in the analysis, as well as boxplots showing the differences in gene expression at the single gene-level. Another aspect that I think could be improved is the presentation of the results in the text, especially in the “Results” section: I think that adding a sentence at the end of each paragraph with a summary of the main observation in the biological context that is then explained in the “Discussion” could help in highlighting the importance of each analysis. In addition, the selection of DNA repair genes should be deeper explained and introduced, as the related paragraph represents one of the most highlighted results. Regarding the “Results” section, the last thing that seemed out of place to me was the logical disposition of the different paragraphs: in particular, I think that the most logical order for the paragraphs would be this one: the PCA analysis, the DEG analysis, the GO and KEGG analyses, and finally the paragraph on DNA repair genes. I think that this type of order would benefit in maintaining the story fluid. Finally, I have a minor revision, regarding the figures: I would increase their resolution, as it is difficult to read the information contained in them.

6. PLOS authors have the option to publish the peer review history of their article (what does this mean?). If published, this will include your full peer review and any attached files.

Reviewer #1: No

---

## [Author Response · Author response to Decision Letter 0]

29 Jul 2024

PONE-D-23-41180

Life in the fastlane? A comparative analysis of gene expression profiles across annual, semi-annual, and non-annual killifishes (Cyprinodontiformes:Nothobranchiidae)

PLOS ONE

Journal Requirements:

KRP - Research reported in this publication was supported by an Institutional Development Award (IDeA) from the National Institute of General Medical Sciences of the National Institutes of Health under grant number P2O GM103424-21

The funders did not play any role in the study design, data collection and analysis, decision to publish, or preparation of the manuscript.

This study was supported by funding to KRP from the Louisiana Biomedical Research Network. Specifically, the research reported in this publication was supported by an Institutional Development Award (IDeA) from the National Institute of General Medical Sciences of the National Institutes of Health under grant number P2O GM103424-21.

KRP - Research reported in this publication was supported by an Institutional Development Award (IDeA) from the National Institute of General Medical Sciences of the National Institutes of Health under grant number P2O GM103424-21

The funders did not play any role in the study design, data collection and analysis, decision to publish, or preparation of the manuscript.

Reviewers' comments:

Reviewer's Responses to Questions

Comments to the Author

1. Is the manuscript technically sound, and do the data support the conclusions?

Reviewer #1: Yes

2. Has the statistical analysis been performed appropriately and rigorously?

Reviewer #1: Yes

3. Have the authors made all data underlying the findings in their manuscript fully available?

Reviewer #1: Yes

4. Is the manuscript presented in an intelligible fashion and written in standard English?

Reviewer #1: Yes

5. Review Comments to the Author

Reviewer #1: The manuscript submitted by Leow and Piller is a descriptive genomic analysis of gene expression profiles from 62 individuals belonging to eleven different Nothobranchiidae species that fall into three different life-histories (annual, semi-annual, and non-annual). The work provides some new information on how gene regulation can change with life-history, with the possibility of studying the genes highlighted in the manuscript in transgenic and gene silencing studies and examining their impact on longevity. 

Reviewer #1: The manuscript submitted by Leow and Piller is a descriptive genomic analysis of gene expression profiles from 62 individuals belonging to eleven different Nothobranchiidae species that fall into three different life-histories (annual, semi-annual, and non-annual). The work provides some new information on how gene regulation can change with life-history, with the possibility of studying the genes highlighted in the manuscript in transgenic and gene silencing studies and examining their impact on longevity. However, it shows also some flaws, the first one being the lack of a deeper discussion on some of the main results, such as the Gene Ontology analysis, as well as the regulation of DNA repair genes. I think that this could be achieved in different ways: creating some heatmaps or pathway visualization to take a further look at the pathways involved in the analysis, as well as boxplots showing the differences in gene expression at the single gene-level. 

We have a DNA repair heatmap (Fig 5) and also have added a subset of single gene level expression boxplots to show the differences in read counts, but the bulk of the DNA repair gene data presented in Tables 2-4.

Another aspect that I think could be improved is the presentation of the results in the text, especially in the “Results” section: I think that adding a sentence at the end of each paragraph with a summary of the main observation in the biological context that is then explained in the “Discussion” could help in highlighting the importance of each analysis. 

We have added summary sentences to several of the paragraphs in the results section, but have refrained from going into great detail. 

In addition, the selection of DNA repair genes should be deeper explained and introduced, as the related paragraph represents one of the most highlighted results. 

We have added a paragraph in the introduction about DNA repair genes and the McCrae et al paper. Also, we have added a few sentences to the methods section to justify our focus on the DNA repair genes.

Regarding the “Results” section, the last thing that seemed out of place to me was the logical disposition of the different paragraphs: in particular, I think that the most logical order for the paragraphs would be this one: the PCA analysis, the DEG analysis, the GO and KEGG analyses, and finally the paragraph on DNA repair genes. I think that this type of order would benefit in maintaining the story fluid. 

 Done.

Finally, I have a minor revision, regarding the figures: I would increase their resolution, as it is difficult to read the information contained in them.

We uploaded the figures into Adobe Photoshop and increased their resolution and re-uploaded them.

---

## [Editor Report · Decision Letter 1]

1 Aug 2024

Life in the fastlane? A comparative analysis of gene expression profiles across annual, semi-annual, and non-annual killifishes (Cyprinodontiformes:Nothobranchiidae)

PONE-D-23-41180R1

Dear Dr. Piller,

We’re pleased to inform you that your manuscript has been judged scientifically suitable for publication and will be formally accepted for publication once it meets all outstanding technical requirements.

Kind regards,

Livia D'Angelo, Ph.D

Academic Editor

PLOS ONE
---

## [Editor Report · Acceptance letter]

29 Aug 2024

PONE-D-23-41180R1 

PLOS ONE

Dear Dr. Piller, 

I'm pleased to inform you that your manuscript has been deemed suitable for publication in PLOS ONE. Congratulations! Your manuscript is now being handed over to our production team.

Kind regards, 

on behalf of

Prof. Livia D'Angelo 

Academic Editor

PLOS ONE